# Body muscle gain and markers of cardiovascular disease susceptibility in young adulthood: A cohort study

Joshua A. Bell[1,2]*, Kaitlin H. Wade[1,2], Linda M. O'Keeffe[1,2,3], David Carslake[1,2], Emma E. Vincent[1,2,4], Michael V. Holmes[1,2,5,6,7], Nicholas J. Timpson[1,2], George Davey Smith[1,2]

1 MRC Integrative Epidemiology Unit, University of Bristol, Bristol, United Kingdom, 2 Population Health Sciences, Bristol Medical School, University of Bristol, Bristol, United Kingdom, 3 School of Public Health, University College Cork, Cork, Ireland, 4 School of Cellular and Molecular Medicine, University of Bristol, Bristol, United Kingdom, 5 Clinical Trial Service Unit & Epidemiological Studies Unit, Nuffield Department of Population Health, University of Oxford, Oxford, United Kingdom, 6 MRC Population Health Research Unit, University of Oxford, Oxford, United Kingdom, 7 National Institute for Health Research Oxford Biomedical Research Centre, Oxford University Hospital, Oxford, United Kingdom

* j.bell@bristol.ac.uk

**Data Availability Statement:** The informed consent obtained from ALSPAC participants does not allow the data to be made freely available through any third party maintained public

## Abstract

### Background

The potential benefits of gaining body muscle for cardiovascular disease (CVD) susceptibility, and how these compare with the potential harms of gaining body fat, are unknown. We compared associations of early life changes in body lean mass and handgrip strength versus body fat mass with atherogenic traits measured in young adulthood.

### Methods and findings

Data were from 3,227 offspring of the Avon Longitudinal Study of Parents and Children (39% male; recruited in 1991–1992). Limb lean and total fat mass indices (kg/m$^2$) were measured using dual-energy X-ray absorptiometry scans performed at age 10, 13, 18, and 25 y (across clinics occurring from 2001–2003 to 2015–2017). Handgrip strength was measured at 12 and 25 y, expressed as maximum grip (kg or lb/in$^2$) and relative grip (maximum grip/weight in kilograms). Linear regression models were used to examine associations of change in standardised measures of these exposures across different stages of body development with 228 cardiometabolic traits measured at age 25 y including blood pressure, fasting insulin, and metabolomics-derived apolipoprotein B lipids. SD-unit gain in limb lean mass index from 10 to 25 y was positively associated with atherogenic traits including very-low-density lipoprotein (VLDL) triglycerides. This pattern was limited to lean gain in legs, whereas lean gain in arms was inversely associated with traits including VLDL triglycerides, insulin, and glycoprotein acetyls, and was also positively associated with creatinine (a muscle product and positive control). Furthermore, this pattern for arm lean mass index was specific to SD-unit gains occurring between 13 and 18 y, e.g., −0.13 SD (95% CI −0.22, −0.04) for VLDL triglycerides. Changes in maximum and relative grip from 12 to 25 y were both

repository. However, data used for this submission can be made available on request to the ALSPAC Executive. The ALSPAC data management plan describes in detail the policy regarding data sharing, which is through a system of managed open access. Full instructions for applying for data access can be found here: http://www.bristol.ac.uk/alspac/researchers/access.

**Funding:** The UK Medical Research Council (MRC), Wellcome (217065/Z/19/Z), and the University of Bristol provide core support for ALSPAC. A comprehensive list of grants funding is available on the ALSPAC website (http://www.bristol.ac.uk/alspac/external/documents/grant-acknowledgements.pdf); this research (involving DXA, handgrip and NMR metabolomics data) was specifically funded by the Wellcome Trust and MRC (076467/Z/05/Z), British Heart Foundation (BHF) (CS/15/6/31468), and MRC (MC_UU_12013/1). JAB and KHW are supported by the Elizabeth Blackwell Institute for Health Research, University of Bristol and the Wellcome Trust Institutional Strategic Support Fund (204813/Z/16/Z). LMOK is supported by a Health Research Board (HRB) of Ireland Emerging Investigator Award (EIA-FA-2019-007 SCaRLeT). EEV is supported by Diabetes UK (17/0005587) and the World Cancer Research Fund (WCRF UK), as part of the World Cancer Research Fund International grant programme (IIG_2019_2009). MVH works in a unit that receives funding from the UK MRC and is supported by a BHF Intermediate Clinical Research Fellowship (FS/18/23/33512) and the National Institute for Health Research Oxford Biomedical Research Centre. NJT is a Wellcome Trust Investigator (202802/Z/16/Z), is the PI of the Avon Longitudinal Study of Parents and Children (MRC & WT 217065/Z/19/Z), is supported by the University of Bristol NIHR Biomedical Research Centre (BRC-1215-20011), the MRC Integrative Epidemiology Unit (MC_UU_12013/3) and works within the CRUK Integrative Cancer Epidemiology Programme (C18281/A29019). JAB, GDS, and DC work in a unit funded by the UK MRC (MC_UU_00011/1) and the University of Bristol. This publication is the work of the authors who are guarantors for its contents. The funders had no role in study design, data collection and analysis, decision to publish, or preparation of the manuscript.

**Competing interests:** MVH has collaborated with Boehringer Ingelheim in research, and in adherence to the University of Oxford's Clinical Trial Service Unit & Epidemiological Studies Unit (CSTU) staff policy, did not accept personal honoraria or other payments from pharmaceutical

positively associated with creatinine, but only change in relative grip was also inversely associated with atherogenic traits, e.g., −0.12 SD (95% CI −0.18, −0.06) for VLDL triglycerides per SD-unit gain. Change in fat mass index from 10 to 25 y was more strongly associated with atherogenic traits including VLDL triglycerides, at 0.45 SD (95% CI 0.39, 0.52); these estimates were directionally consistent across sub-periods, with larger effect sizes with more recent gains. Associations of lean, grip, and fat measures with traits were more pronounced among males. Study limitations include potential residual confounding of observational estimates, including by ectopic fat within muscle, and the absence of grip measures in adolescence for estimates of grip change over sub-periods.

## Conclusions

In this study, we found that muscle strengthening, as indicated by grip strength gain, was weakly associated with lower atherogenic trait levels in young adulthood, at a smaller magnitude than unfavourable associations of fat mass gain. Associations of muscle mass gain with such traits appear to be smaller and limited to gains occurring in adolescence. These results suggest that body muscle is less robustly associated with markers of CVD susceptibility than body fat and may therefore be a lower-priority intervention target.

## Author summary

### Why was this study done?

- Higher body fat likely causes heart disease, but fat loss remains difficult to maintain. Evidence is less robust on whether gaining body muscle mass or strength would reduce the risk of heart disease, and how the size of potential benefit from muscle or strength gain compares with the expected harm of fat gain.

- Examining naturally occurring changes in lean mass, grip strength, and fat mass across early stages of life, when ageing-related chronic diseases are rare, should naturally reduce the potential for confounding by subclinical disease and enable less biased estimates of the effect of each body compartment on markers of heart health.

### What did the researchers do and find?

- We used data on approximately 3,000 young people from a British birth cohort study to examine repeated measures of body fat and lean mass taken from body scanning performed during childhood, adolescence, and young adulthood, as well as repeated measures of handgrip strength from childhood and young adulthood.

- We examined associations between these exposures and detailed measures taken from blood samples in young adulthood including apolipoprotein-B-related cholesterol, which reflects susceptibility to heart disease. This enabled us to compare how strongly different body compartments relate to heart health and to pinpoint at what stage of early life (before adulthood) each may be most impactful.

companies. GDS is a PLOS Medicine editorial board member. No others to declare.

**Abbreviations:** ALSPAC, Avon Longitudinal Study of Parents and Children; BMI, body mass index; CHD, coronary heart disease; CVD, cardiovascular disease; DBP, diastolic blood pressure; DXA, dual-energy X-ray absorptiometry; GlycA, glycoprotein acetyls; MR, Mendelian randomisation; non-HDL, non-high-density lipoprotein; RCT, randomised controlled trial; SBP, systolic blood pressure; VLDL, very-low-density lipoprotein; Short title, Body muscle and CVD susceptibility.

- We found that gaining lean mass and grip strength were only weakly related to healthier levels of blood markers in young adulthood, and mainly among males, with only lean mass gains occurring in adolescence appearing potentially beneficial. Gaining fat mass was more strongly and consistently related to poorer health in young adulthood, again particularly among males.

## What do these findings mean?

- These findings suggest that greater benefits to heart health may be expected from reducing body fat than from gaining body muscle. They further suggest that the regular use of muscle matters more than the volume or intentional building up of muscle for avoiding heart disease.

- Body muscle is still likely to benefit other functional aspects of health including mobility, and these benefits should still be relayed to patients and the public.

## Introduction

Cardiovascular diseases (CVDs) remain leading causes of early mortality [1]. Multiple lines of evidence from population and mechanistic studies support higher body fat as a likely cause of such diseases, including coronary heart disease (CHD) [2–5]. These harms of body fatness are thought to be driven largely by its effects on cardiometabolic intermediates including higher blood pressure, apolipoprotein-B-containing lipoproteins, and glucose [6,7]. Population reductions in body fat remain difficult to achieve, however [8]. This reality motivates the direct targeting of intermediate traits and of other metabolically active, and potentially modifiable, body tissues.

Body muscle is metabolically active and its contraction is expected to be anti-inflammatory and anti-hyperglycaemic [9]. Higher total lean mass has shown adverse cardiometabolic profiles [10,11] however, possibly reflecting residual confounding by fat in abdominal regions [4]. Lean mass held within limbs may better isolate skeletal muscle as these compartments correlate most highly with muscle volume measured by magnetic resonance imaging [12,13]. Benefits of muscle may also be reflected in strength, which can be measured directly for isolated arm muscles using handgrip tests; grip strength correlates well (>0.7) with objectively measured strength in other muscle groups such as hips and is thus a useful and widely used proxy for overall muscular strength in larger scale studies [14–16]. Prospective observational estimates suggest that higher limb lean mass and stronger grip are both associated with lower CHD risk independent of body mass index (BMI) [14,17,18], and factorial Mendelian randomisation (MR) estimates suggest that the risk ratio for CVD onset is comparable among adults with high grip strength and high BMI (1.04; 95% CI 0.98, 1.11) and among adults with low grip strength and high BMI (1.03; 95% CI 0.97, 1.10), compared with adults with high grip strength and low BMI (*P* value for interaction = 0.50) [19]. Such factorial MR estimates for limb lean mass, whether based on bioimpedance or more precise dual-energy X-ray absorptiometry (DXA) scans, are not yet available.

The likely causality between body muscle and CVD susceptibility can be interrogated by examining the association of body muscle with intermediate traits that have triangulated

evidence of causality for CVD in samples that have reduced potential for confounding. Higher limb lean mass and stronger grip are both associated with lower glycaemia, apolipoprotein B lipids, blood pressure, and inflammation [20–23]. These associations for grip appear stronger when expressed as a function of, rather than adjusting for, weight or BMI [20–25]. Notably, estimates for limb lean mass and grip are based largely on middle- to older-age adults, which limits causal inference given the high potential for confounding by subclinical disease (reverse causation). The few studies of children or young adults suggest weak associations of stronger grip with total cholesterol, glucose, and blood pressure [26–29], and potentially positive associations of higher limb lean mass with atherogenic lipids, glycaemia, and blood pressure [11]. Measuring muscle and strength earlier in life, when subclinical diseases are rare, should enable less biased estimates of cardiometabolic effects.

Growing evidence suggests that associations between body fat mass and atherogenic traits differ importantly by sex. Higher fat mass, whether measured using BMI, waist circumference, or DXA fat mass, appears to be more strongly positively associated among males with CVD-relevant traits including metabolomics-derived glucose, apolipoprotein B lipids, and inflammatory glycoprotein acetyls (GlycA) [7,30]; these sex differences in turn vary by stage of body development, with associations appearing more adverse among males in childhood, adolescence, and young adulthood, but similar or more adverse among females in middle adulthood [30]. Such differences do not simply reflect differences in total fat volume since males tend to carry less total fat than females throughout life [30,31]; these differences instead suggest an important role of abdominal and ectopic fat storage, which is often higher among males [4,31], in underpinning the effects of total fat [10]. Males tend to have higher lean mass and stronger grip than females [14,32], but whether important sex differences exist in the associations of muscle mass or strength with atherogenic traits is unknown. Sex differences in the potential benefits of muscle have also not been previously examined in relation to detailed atherogenic traits measured from targeted metabolomics [33]. Muscle tissue is thought to be more modifiable after childhood [34], and thus greater benefits from a higher contractile capacity may be expected from adolescence onwards, but the common lack of repeated measures of body muscle at different life stages has prevented examination of the potential modifying role of growth and development in the associations of muscle change with atherogenic traits [35].

We aimed in this study to estimate the effects of gaining body muscle on markers of CVD susceptibility using repeated measures of DXA limb lean mass and grip strength across early life in relation to blood pressure and metabolomics-derived atherogenic traits in young adulthood. We examined associations of change in limb lean mass and grip from childhood to young adulthood with atherogenic traits, and whether associations differ by stage of body development and by sex. We examined associations of change in DXA fat mass index with traits in the same manner, to directly compare the magnitude of potential benefits of gaining muscle with the potential harms of gaining fat.

## Methods

### Study population

Data were from Generation 1 of the Avon Longitudinal Study of Parents and Children (ALSPAC), a population-based birth cohort study in which 15,454 pregnant women (Generation 0) with an expected delivery date between 1 April 1991 and 31 December 1992 were recruited from the former Avon County of southwest England [36]. Since then, 14,901 Generation 1 individuals alive at 1 y have been followed repeatedly with questionnaire- and clinic-based assessments [37–39], including an additional 913 Generation 1 individuals enrolled over the course of the study [40]. Written informed consent was provided, and ethical approval was

obtained from the ALSPAC Ethics and Law Committee and the local research ethics committee. Consent for biological samples was collected in accordance with the UK Human Tissue Act 2004. Written informed consent for the use of data collected via questionnaires and clinics was obtained from participants following recommendations of the ALSPAC Ethics and Law Committee at the time. The study website contains details of all available data through a fully searchable data dictionary and variable search tool (http://www.bristol.ac.uk/alspac/researchers/our-data/).

Our study aims, objectives, and analytical intentions were summarised in March 2019 prior to data handling for the purposes of an ALSPAC data application (S1 Study Plan). This study was initially motivated by metabolomics work on type 2 diabetes susceptibility [41] and was expanded to investigate CVD susceptibility more broadly. Examinations of sex differences were motivated by more recent work on body fatness [30] and prior peer review recommendations. This study is reported as per the Strengthening the Reporting of Observational Studies in Epidemiology (STROBE) guidelines (S1 STROBE Checklist).

We conducted main analyses on unrestricted samples of participants (with $N$ varying between traits and across occasions) to enable use of all measured data. Of the 14,901 surviving Generation 1 ALSPAC participants who were eligible for future clinic assessments, 6,119 participants had data on covariates used for model adjustments plus $\geq 1$ of any of the following: DXA measures (lean or fat mass) or grip strength at age 25 y; change in DXA measures across the total observation period (10 to 25 y), across childhood (10 to 13 y), across adolescence (13 to 18 y), or across young adulthood (18 to 25 y); or change in grip strength across the total observation period (12 to 25 y). Of those 6,119 participants with any exposure measure, 3,227 participants had data on $\geq 1$ cardiometabolic trait at age 25 y and were thus considered eligible for inclusion in at least 1 of the present set of analyses (Fig 1).

## Assessing body muscle mass and strength

When aged approximately 10, 12, 13, 18, and 25 y, participants underwent body scanning using a DXA Lunar Prodigy narrow fan beam densitometer, from which total and regional lean mass (in kilograms, excluding fat and bone) was estimated. Scans were screened for anomalies, motion, and material artefacts, and realigned when necessary [42]. Limb lean mass was calculated by summing lean mass in arms and legs (trunk excluded). On each occasion, height was measured in light clothing without shoes to the nearest 0.1 cm using a Harpenden stadiometer. Limb lean mass index was calculated using squared height (kg/m$^2$). Separate lean mass indices for arms and legs were also calculated. Fat mass index was calculated based on total body fat mass (kg/m$^2$).

Participants underwent handgrip strength testing on 2 occasions: when aged approximately 12 y using a Jamar hydraulic dynamometer and when aged approximately 25 y using a Baseline pneumatic squeeze bulb dynamometer. The latter device records grip more dynamically, with units scaled to hand size; previous studies of younger and older adults suggest that measurements from squeeze bulb dynamometers and Jamar hydraulic devices are highly correlated ($r > 0.8$) and similarly detect sex differences in grip strength [43,44]. On each occasion, participants sat in a chair with arms and back supported and were asked to rest their forearms on the arms of the chair, with their wrist just over the end of the chair arm (thumb facing upwards, with wrist in a neutral position). Participants were asked to squeeze the device as tightly and for as long as possible, 3 times in succession using their dominant (writing) hand, to record maximum isometric strength (in kilograms of force at 12 y and in pounds per square inch of force at 25 y). Maximum grip strength was estimated as the mean of 3 measures. Grip strength as a function of body weight, here termed 'relative grip', was calculated as maximum grip divided by body weight in kilograms. We scaled for weight rather than fat mass index or BMI

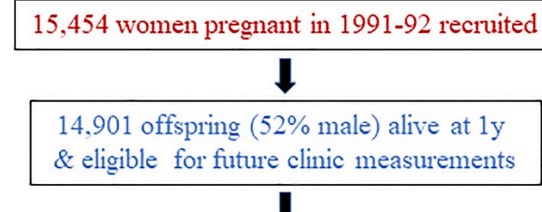

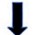

**Fig 1. Selection of Avon Longitudinal Study of Parents and Children Generation 1 participants eligible for inclusion in ≥1 analysis.** DXA, dual-energy X-ray absorptiometry.

when deriving relative grip following peer review; this approach was considered preferable because it (1) accounts for overall body size rather than adiposity specifically, (2) reduces bias from an induced negative correlation between relative grip strength and adiposity, (3) enables consistent multivariable adjustment for fat mass index as a confounder across muscle measures, and (4) enables comparable estimates of effect with MR analyses, given that genome-wide association studies have scaled for weight rather than BMI [45].

## Assessing cardiometabolic traits

When the participants were aged approximately 25 y, systolic blood pressure (SBP) and diastolic blood pressure (DBP) were examined twice in succession while the individual was seated with the arm supported, using an appropriately sized cuff and a DINAMAP 9301 device. Mean levels of each were used to represent resting SBP and DBP. Fasting blood samples were drawn, from which insulin (mu/l) and C-reactive protein (CRP) (mg/l) were quantified using routine clinical chemistry. Proton nuclear magnetic resonance (NMR) spectroscopy from targeted metabolomics [33] was also performed to quantify 145 concentrations (mostly mmol/l) and 79 ratios describing traits including cholesterol and triglyceride content of lipoprotein subclasses, apolipoprotein B, glucose, branched chain amino acids (BCAAs), creatinine (a muscle product and positive control), and inflammatory GlycA.

## Assessing confounders

The measured confounders included sex, ethnicity (white versus non-white), age at the time of exposure (lean mass, grip, fat mass) assessment, and highest level of education attained by the participant's mother as reported shortly after delivery (certificate of secondary education, vocational, O-level, A-level, or degree, using English standards) to indicate socioeconomic position at birth. Smoking at age 18 y and 25 y was recorded via questionnaire and grouped as never smoked an entire cigarette, smokes less than weekly, or smokes weekly. Alcohol consumption at 18 y and 25 y was recorded and grouped as never/monthly/less than monthly, 2–4

times/month, or ≥2 times/week. Puberty timing was estimated as age at peak height velocity based on SuperImposition by Translation and Rotation (SITAR) modelling of heights from 5 y to 20 y (detailed previously [46]).

## Analyses

Pearson correlation coefficients were examined between changes in lean and fat mass indices based on age 10 and 25 y measures, and between changes in grip and lean and fat mass indices based on age 12 and 25 y measures. Correlations were also examined between changes in lean and fat mass indices across sub-periods of childhood (10 to 13 y), adolescence (13 to 18 y), and young adulthood (18 to 25 y).

Exposures and outcomes were analysed in standardised *z*-score units to allow comparability of effect sizes given dissimilar variances between traits and across occasions (this also improved comparability between handgrip strength measures, given the different units across occasions). Because exposure distributions differed substantially by sex (S1–S5 Figs), as did exposure change distributions (S6–S10 Figs), exposures were *z*-scored separately within each sex. We then examined associations of change in lean mass indices (total limb, arm, and leg) based on difference scores (standardised index at 25 y minus standardised index at 10 y) with cardiometabolic traits at 25 y using linear regression models with robust standard errors. Models adjusted for age, sex, ethnicity, maternal education, the 10-y value of the index being assessed, change in the other lean mass index (other limb compartment), and change in fat mass index. We examined whether association patterns differed by stage of body development by repeating analyses based on change in lean mass indices from 10 to 13 y (childhood), 13 to 18 y (adolescence), and 18 to 25 y (young adulthood), in relation to traits at 25 y. Models of change in adolescence were additionally adjusted for age at peak height velocity, while models of change in young adulthood were additionally adjusted for this plus smoking and alcohol at 18 y. Associations of change in grip measures from 12 to 25 y with traits at 25 y were examined with adjustment for basic demographic factors, grip strength at 12 y, and change in fat mass index.

We examined associations of change in fat mass index from 10 to 25 y, and over sub-periods, with traits at 25 y, using the same model adjustment strategies as for limb lean mass index but adjusting for limb lean (instead of fat) mass index. Lastly, following peer review, we examined cross-sectional associations of lean mass indices, grip strength, and fat mass index with cardiometabolic traits, all measured at 25 y, to compare the association profile of changes in exposures to that of current levels of exposure. These models were adjusted for the same covariates as in models with exposure change measured starting from 18 y, but with smoking, alcohol, and lean/fat mass indices now measured at 25 y. We examined whether associations differ substantially by sex by repeating all analyses among males and females separately.

Since main analyses were conducted on unrestricted samples of participants (with *N* varying between traits and across occasions), we repeated models using 770 participants with data on every DXA and grip measure at every time point, every cardiometabolic trait, and every covariate (complete case), to examine whether results are sensitive to changing sample size.

As recommended for aims of estimation [47,48], we present exact *P* values and base our interpretations of results on effect size and precision. Analyses were done using Stata 15.1 (StataCorp, College Station, Texas, US).

## Results

### Sample characteristics

In total, 3,227 participants contributed to analyses (39% male) (Table 1; Fig 1). Mean (SD) age at peak height velocity was 12.4 y (1.2 y) overall (range: 9.1 y to 17.4 y). Based on this indicator,

**Table 1. Characteristics of 3,227 Avon Longitudinal Study of Parents and Children Generation 1 offspring eligible for analyses.**

| Characteristics | Overall (N = 3,227) | | Males (N = 1,257) | | Females (N = 1,970) | |
|---|---|---|---|---|---|---|
| | N | Percent (N) or mean (SD) | N | Percent (N) or mean (SD) | N | Percent (N) or mean (SD) |
| Non-white ethnicity | 3,227 | 3.9% (126) | 1,257 | 3.7% (47) | 1,970 | 4.0% (79) |
| Maternal education is degree | 3,227 | 21.0% (676) | 1,257 | 22.4% (282) | 1,970 | 20.0% (394) |
| Age (y) at peak height velocity | 2,924 | 12.4 (1.2) | 1,155 | 13.5 (0.9) | 1,769 | 11.8 (0.8) |
| Smoking at 25 y | 3,191 | | 1,240 | | 1,951 | |
| Never | | 36.1% (1,153) | | 35.4% (439) | | 36.6% (714) |
| Less than weekly | | 58.3% (1,859) | | 57.6% (714) | | 58.7% (1,145) |
| Every week | | 5.6% (179) | | 7.0% (87) | | 4.7% (92) |
| Alcohol consumption at 25 y | 3,123 | | 1,209 | | 1,914 | |
| Never/monthly/less than monthly | | 24.0% (748) | | 17.6% (213) | | 28.0% (535) |
| 2 to 4 times per month | | 38.5% (1,202) | | 36.6% (443) | | 39.7% (759) |
| 2 or more times per week | | 37.6% (1,173) | | 45.7% (553) | | 32.4% (620) |
| Total limb lean mass index (kg/m$^2$) at 25 y | 3,119 | 7.2 (1.3) | 1,222 | 8.2 (1.1) | 1,897 | 6.5 (0.9) |
| Arm lean mass index (kg/m$^2$) at 25 y | 3,119 | 1.7 (0.5) | 1,222 | 2.2 (0.4) | 1,897 | 1.5 (0.2) |
| Leg lean mass index (kg/m$^2$) at 25 y | 3,119 | 5.4 (0.9) | 1,222 | 6.0 (0.8) | 1,897 | 5.1 (0.7) |
| Maximum grip strength (lb/in$^2$) at 25 y | 1,964 | 14.1 (3.9) | 771 | 17.4 (3.7) | 1,193 | 12.1 (2.3) |
| Relative grip strength (lb/in$^2$/kg) at 25 y | 1,951 | 0.2 (0.1) | 768 | 0.2 (0.1) | 1,183 | 0.2 (0.04) |
| Total fat mass index (kg/m$^2$) at 25 y | 3,119 | 7.9 (3.7) | 1,222 | 6.3 (3.0) | 1,897 | 9.0 (3.8) |
| **Changes from childhood to young adulthood, 10 to 25 y** | | | | | | |
| Total limb lean mass index (kg/m$^2$) | 2,808 | 1.8 (1.0) | 1,102 | 2.6 (0.9) | 1,706 | 1.3 (0.7) |
| Arm lean mass index (kg/m$^2$) | 2,808 | 0.6 (0.4) | 1,102 | 0.9 (0.3) | 1,706 | 0.3 (0.2) |
| Leg lean mass index (kg/m$^2$) | 2,808 | 1.2 (0.7) | 1,102 | 1.6 (0.7) | 1,706 | 1.0 (0.6) |
| Maximum grip strength (SD)* | 1,739 | −0.001 (1.1) | 681 | −0.004 (1.1) | 1,058 | 0.001 (1.1) |
| Relative grip strength (SD)* | 1,727 | −0.04 (1.0) | 678 | −0.01 (1.0) | 1,049 | −0.1 (1.0) |
| Total fat mass index (kg/m$^2$) | 2,808 | 3.6 (3.0) | 1,102 | 2.6 (2.4) | 1,706 | 4.2 (3.1) |
| **Changes in childhood, 10 to 13 y** | | | | | | |
| Total limb lean mass index (kg/m$^2$) | 2,565 | 1.0 (0.6) | 1,017 | 1.3 (0.6) | 1,548 | 0.7 (0.4) |
| Arm lean mass index (kg/m$^2$) | 2,565 | 0.3 (0.2) | 1,017 | 0.4 (0.2) | 1,548 | 0.2 (0.1) |
| Leg lean mass index (kg/m$^2$) | 2,565 | 0.7 (0.4) | 1,017 | 1.0 (0.4) | 1,548 | 0.5 (0.3) |
| Total fat mass index (kg/m$^2$) | 2,565 | 0.9 (1.7) | 1,017 | 0.3 (1.6) | 1,548 | 1.3 (1.6) |
| **Changes in adolescence, 13 to 18 y** | | | | | | |
| Total limb lean mass index (kg/m$^2$) | 2,404 | 0.3 (0.6) | 934 | 0.8 (0.7) | 1,470 | −0.02 (0.4) |
| Arm lean mass index (kg/m$^2$) | 2,404 | 0.2 (0.2) | 934 | 0.4 (0.2) | 1,470 | 0.1 (0.1) |
| Leg lean mass index (kg/m$^2$) | 2,404 | 0.1 (0.5) | 934 | 0.3 (0.5) | 1,470 | −0.1 (0.3) |
| Total fat mass index (kg/m$^2$) | 2,404 | 1.1 (2.0) | 934 | 0.3 (1.9) | 1,470 | 1.6 (1.9) |
| **Changes in young adulthood, 18 to 25 y** | | | | | | |
| Total limb lean mass index (kg/m$^2$) | 2,543 | 0.5 (0.7) | 983 | 0.4 (0.8) | 1,560 | 0.6 (0.7) |
| Arm lean mass index (kg/m$^2$) | 2,543 | 0.1 (0.2) | 983 | 0.1 (0.3) | 1,560 | 0.03 (0.2) |
| Leg lean mass index (kg/m$^2$) | 2,543 | 0.5 (0.6) | 983 | 0.3 (0.6) | 1,560 | 0.6 (0.6) |
| Total fat mass index (kg/m$^2$) | 2,543 | 1.6 (2.4) | 983 | 2.2 (2.1) | 1,560 | 1.3 (2.5) |

Described are those with data on change in at least 1 lean, grip, or fat measure across any occasion and covariates used for those models, and at least 1 cardiometabolic trait at 25 y.

*Change is from age 12 y to 25 y and is based on difference in SD units, given different original measurement units between occasions.

0.4% of participants entered puberty by 10 y (when lean/fat mass was first assessed), 29.9% did by 12 y (when grip was first assessed), and all did by 25 y. Changes in lean mass indices were generally positive but smaller than positive changes in fat mass index; all were highly variable

(Table 1). Overall, males experienced more positive change in lean mass indices and grip, whereas females experienced more positive change in fat mass index; sex differences appeared largest before young adulthood (Table 1; S6–S10 Figs).

Ineligible participants were more likely than eligible participants to be male (54.5%) and to have lower maternal education; they also showed similar smoking patterns but lower levels of weekly drinking at 25 y (S1 Table). Changes in lean and fat mass indices were similar.

## Correlations among lean mass indices, grip strength, and fat mass index

Change in limb lean mass index between ages 12 and 25 y was positively correlated with change in maximum grip ($r$ = 0.33) but negatively correlated with change in relative grip ($r$ = −0.12) over this same period. Change in limb lean mass index was positively correlated with change in fat mass index over this same period ($r$ = 0.39), as well as between the ages 10 and 25 y ($r$ = 0.47). Change in limb lean mass index was more positively correlated with change in leg lean mass index than change in arm lean mass index, based on either time period (S2 and S3 Tables). Based on changes occurring between the ages 12 and 25 y, fat mass index change was uncorrelated with maximum grip change ($r$ = 0.03) but negatively correlated with relative grip change ($r$ = −0.44). Considering different sub-periods, change in fat mass index was more positively correlated with change in arm lean mass index than change in leg lean mass index over childhood and adolescence (S4 and S5 Tables), but was more positively correlated with change in leg than arm lean mass index in young adulthood (S6 Table). A similar correlation pattern was seen cross-sectionally based on measures taken at age 25 y (S7 Table).

## Associations of changes in limb lean mass indices with cardiometabolic traits

Evidence was strong for associations of change in limb lean mass index from 10 to 25 y (per SD-unit gain) with higher creatinine and most atherogenic traits; these were of modest magnitude and mostly in directions assumed to reflect poorer health, e.g., 0.17 SD (95% CI 0.10, 0.24) higher very-low-density lipoprotein (VLDL) triglycerides (Fig 2; S8 Table). When examining gains in arm and leg lean mass indices separately, only gain in arm lean mass was positively associated with creatinine. The adverse pattern of associations seen across traits with limb lean mass index gain appeared limited to gain in leg lean mass, whereas gain in arm lean mass was associated with traits including lower VLDL triglycerides (−0.09 SD; 95% CI −0.15, −0.02), insulin, GlycA, and DBP, but higher SBP. This pattern for creatinine and atherogenic traits was more pronounced among males (S9 and S10 Tables).

Change in limb lean mass index from 10 to 13 y was generally associated with higher atherogenic lipids at 25 y including low-density lipoprotein cholesterol and VLDL triglycerides, at 0.19 SD (95% CI 0.13, 0.26); positive associations were apparent with apolipoprotein B, insulin, GlycA, SBP, and DBP (Figs 3 and 4; S11 Table). Gain in both limb compartments was positively associated with creatinine, and adverse trait profiles seen with childhood gain in limb lean mass index were reflected in arms as well as legs. Effect sizes appeared larger among males (S12 and S13 Tables).

Change in limb lean mass index from 13 to 18 y was generally unassociated with creatinine and atherogenic traits at 25 y, apart from weak associations with higher SBP and DBP (Figs 3 and 4; S14 Table). In contrast, gain in arm lean mass index was positively associated with creatinine and inversely associated with atherogenic traits including VLDL triglycerides (−0.13 SD; 95% CI −0.22, −0.04), apolipoprotein B, insulin, GlycA, and DBP, but positively associated with SBP. Associations were again stronger among males (S15 and S16 Tables).

Change in limb lean mass index from 18 to 25 y was associated with higher creatinine and atherogenic traits at 25 y including higher VLDL triglycerides (Figs 3 and 4; S17 Table). The

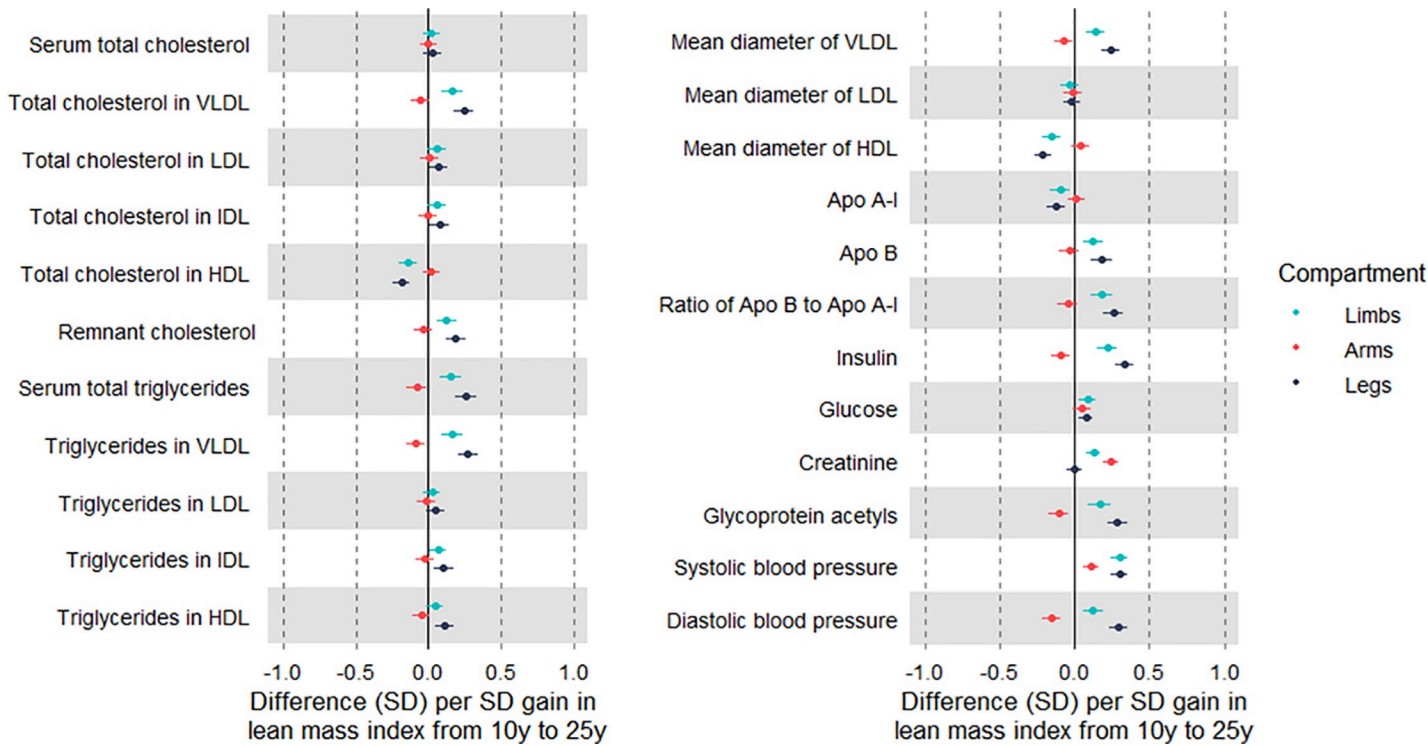

**Fig 2. Associations of change in limb lean mass indices from age 10 y to 25 y with cardiometabolic traits at 25 y among Avon Longitudinal Study of Parents and Children Generation 1 individuals.** Estimates are beta coefficients and 95% CIs representing SD-unit differences in cardiometabolic traits at 25 y per SD-unit gain in lean mass index from 10 y to 25 y (based on standardised index at 25 y minus standardised index at 10 y). Models adjusted for age, sex, ethnicity, maternal education, lean mass index at 10 y, change in the other lean mass index, and change in total fat mass index ($N$ range: 2,121 to 2,804). Limb lean mass index defined as sum of lean mass in arms and legs (kg) divided by squared height (m$^2$). Apo, apolipoprotein; LDL, low-density lipoprotein; HDL, high-density lipoprotein; IDL, intermediate-density lipoprotein; VLDL, very-low-density lipoprotein.

positive association with creatinine was again exclusive to arms, and gain in arm lean mass was generally unassociated with atherogenic traits except for a positive association with DBP. Such null associations were seen among males and females, despite a strong positive association of gain in arm lean mass with creatinine among males (S18 and S19 Tables).

## Associations of change in grip strength with cardiometabolic traits

Change in maximum grip from 12 to 25 y (per SD-unit gain) was positively associated with creatinine, but associations were largely null with atherogenic traits, including VLDL triglycerides (−0.01 SD; 95% CI −0.07, 0.04) (Fig 5; S20 Table), among males and females (S21 and S22 Tables). In contrast, change in relative grip over the same period was positively associated with creatinine and moderately inversely associated with atherogenic traits, e.g., VLDL triglycerides, at −0.12 SD (95% CI −0.18, −0.06), with similar magnitudes for apolipoprotein B, insulin, GlycA, SBP, and DBP. These associations appeared strongest among females.

## Associations of change in fat mass index with cardiometabolic traits

Change in fat mass index (per SD-unit gain) was inversely associated with creatinine and strongly positively associated with atherogenic traits including VLDL triglycerides (0.45 SD; 95% CI 0.39, 0.52), apolipoprotein B, insulin, GlycA, SBP, and DBP (Fig 6; S23 Table). When examining gains over sub-periods in relation to traits at 25 y, estimates were directionally consistent across occasions, with a tendency for larger effect sizes with more recent gains. For

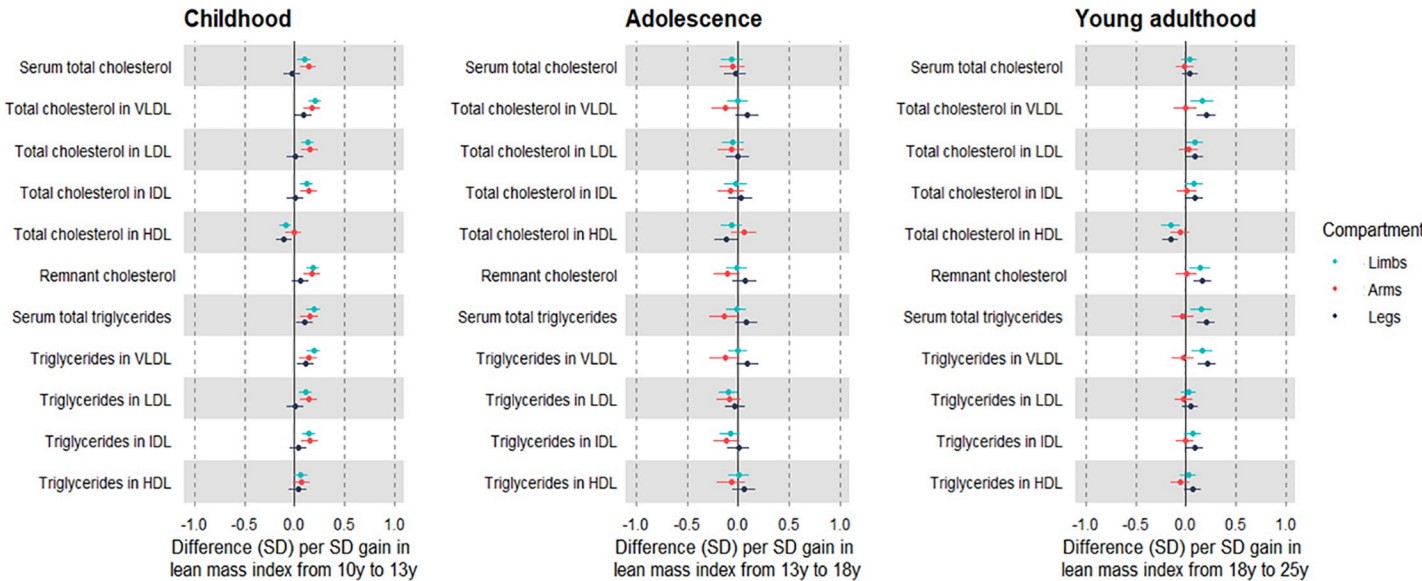

**Fig 3. Associations of change in limb lean mass indices across different life stages with lipid traits at 25 y among Avon Longitudinal Study of Parents and Children Generation 1 individuals.** Estimates are beta coefficients and 95% CIs representing SD-unit differences in cardiometabolic traits at 25 y per SD-unit change in lean mass index (based on standardised index at time 2 minus standardised index at time 1). Childhood models are based on change in lean mass index from 10 y to 13 y and are adjusted for age, sex, ethnicity, maternal education, lean mass index at 10 y, change in the other lean mass index, and change in total fat mass index (N range: 1,926 to 2,557). Adolescence models are based on change in lean mass index from 13 y to 18 y and are adjusted for age, sex, ethnicity, maternal education, puberty timing, change in the other lean mass index, and change in total fat mass index (N range: 1,747 to 2,344). Young adulthood models are based on change in lean mass index from 18 y to 25 y and are adjusted for age, sex, ethnicity, maternal education, puberty timing, smoking, alcohol, change in the other lean mass index, and change in total fat mass index (N range: 1,532 to 2,036). Limb lean mass index defined as sum of lean mass in arms and legs (kg) divided by squared height (m²). LDL, low-density lipoprotein; HDL, high-density lipoprotein; IDL, intermediate-density lipoprotein; VLDL, very-low-density lipoprotein.

example, point estimates for associations of fat gain in childhood, adolescence, and young adulthood with VLDL triglycerides at 25 y were 0.10 SD, 0.34 SD, and 0.48 SD, respectively. Associations were more pronounced among males, at about double the magnitude of effect size, versus females (S24 and S25 Tables).

### Cross-sectional associations of lean mass indices, grip strength, and fat mass index with cardiometabolic traits in young adulthood

Evidence was weaker for cross-sectional associations of higher limb lean mass (per SD) with atherogenic lipid, glycaemic, and inflammatory traits, e.g., −0.04 SD (95% CI −0.10, 0.01) for apolipoprotein B (S26 Table). Also, in contrast to estimates based on change in exposure, estimates appeared more favourable for each SD higher leg lean mass index, e.g., apolipoprotein B was −0.10 SD (95% CI −0.18, −0.01) lower but 0.05 SD (95% CI −0.02, 0.11) higher for leg and arm lean mass indices, respectively. Leg lean mass index was more positively associated with creatinine than was arm lean mass index (point estimates were 0.21 SD and 0.13 SD, respectively). These associations were more pronounced among females than males despite similarly positive associations of lean mass indices with creatinine between the sexes (S27 and S28 Tables).

Evidence was substantially weaker or null for cross-sectional associations of both grip strength measures (per SD) with atherogenic lipid, glycaemic, and inflammatory traits, e.g., 0.01 SD (95% CI −0.05, 0.07) for apolipoprotein B with stronger relative grip (S29 Table). Associations were more evident among males, with estimates appearing positive in relation to VLDL lipids, e.g., 0.10 SD (95% CI −0.02, 0.22) for total lipids in very large VLDL (S30 and

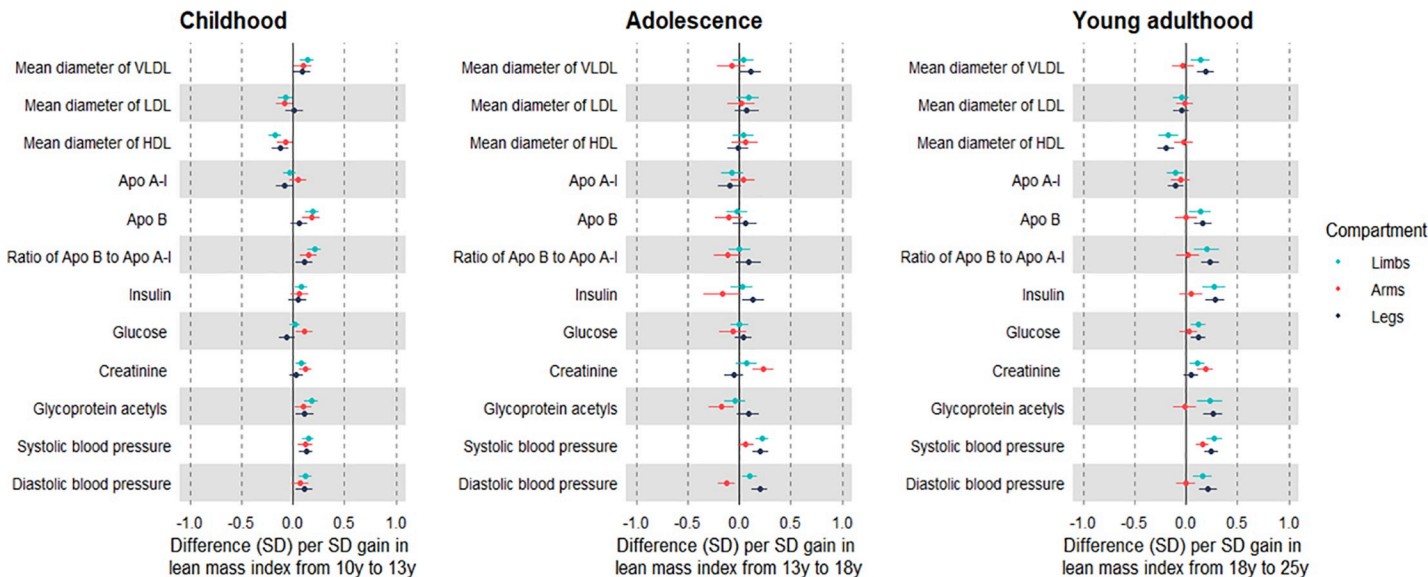

**Fig 4. Associations of change in limb lean mass indices across different life stages with lipid, pre-glycaemic, inflammatory, and blood pressure traits at 25 y among Avon Longitudinal Study of Parents and Children Generation 1 individuals.** Estimates are beta coefficients and 95% CIs representing SD-unit differences in cardiometabolic traits at 25 y per SD-unit change in lean mass index (based on standardised index at time 2 minus standardised index at time 1). Childhood models are based on change in lean mass index from 10 y to 13 y and are adjusted for age, sex, ethnicity, maternal education, lean mass index at 10 y, change in the other lean mass index, and change in total fat mass index (N range: 1,926 to 2,557). Adolescence models are based on change in lean mass index from 13 y to 18 y and are adjusted for age, sex, ethnicity, maternal education, puberty timing, change in the other lean mass index, and change in total fat mass index (N range: 1,747 to 2,344). Young adulthood models are based on change in lean mass index from 18 y to 25 y and are adjusted for age, sex, ethnicity, maternal education, puberty timing, smoking, alcohol, change in the other lean mass index, and change in total fat mass index (N range: 1,532 to 2,036). Limb lean mass index defined as sum of lean mass in arms and legs (kg) divided by squared height (m$^2$). Apo, apolipoprotein; LDL, low-density lipoprotein; HDL, high-density lipoprotein; VLDL, very-low-density lipoprotein.

S31 Tables). Maximum grip was more positively associated with creatinine than was relative grip, in both sexes.

In contrast, the direction and magnitude of cross-sectional associations of higher fat mass index with atherogenic traits were highly comparable to those from previous models of change in exposure (S32 Table). For example, higher fat mass index (per SD) was associated with 0.42 SD (95% CI 0.36, 0.48) higher triglycerides in VLDL, 0.47 SD (95% CI 0.41, 0.53) higher insulin, and 0.53 SD (95% CI 0.48, 0.58) higher GlycA. Associations for most traits were comparable between the sexes, except for non-high-density lipoprotein (non-HDL) lipids, which were more positive among males (S32 Table).

Estimates based on complete case analyses were comparable to those of the main analyses in terms of direction and magnitude, with expectedly lower precision given smaller Ns (S8–S32 Tables).

## Discussion

This study aimed to estimate the potential benefits of gaining body muscle for markers of CVD susceptibility in young adulthood, and how these compare with the potential harms of gaining body fat. We integrated repeated measures of DXA limb lean and fat mass indices and grip strength starting in childhood with metabolomic measures of atherogenic traits taken in young adulthood. Our results suggest that muscle strengthening, as indicated by grip strength gain, is weakly associated with lower atherogenic trait levels, particularly among males. Associations of gain in muscle mass with traits were smaller in magnitude and limited to gains occurring in adolescence. Gaining body fat was more consistently associated with the same traits, in unfavourable directions and at larger magnitudes than seen for muscle mass or strength, again

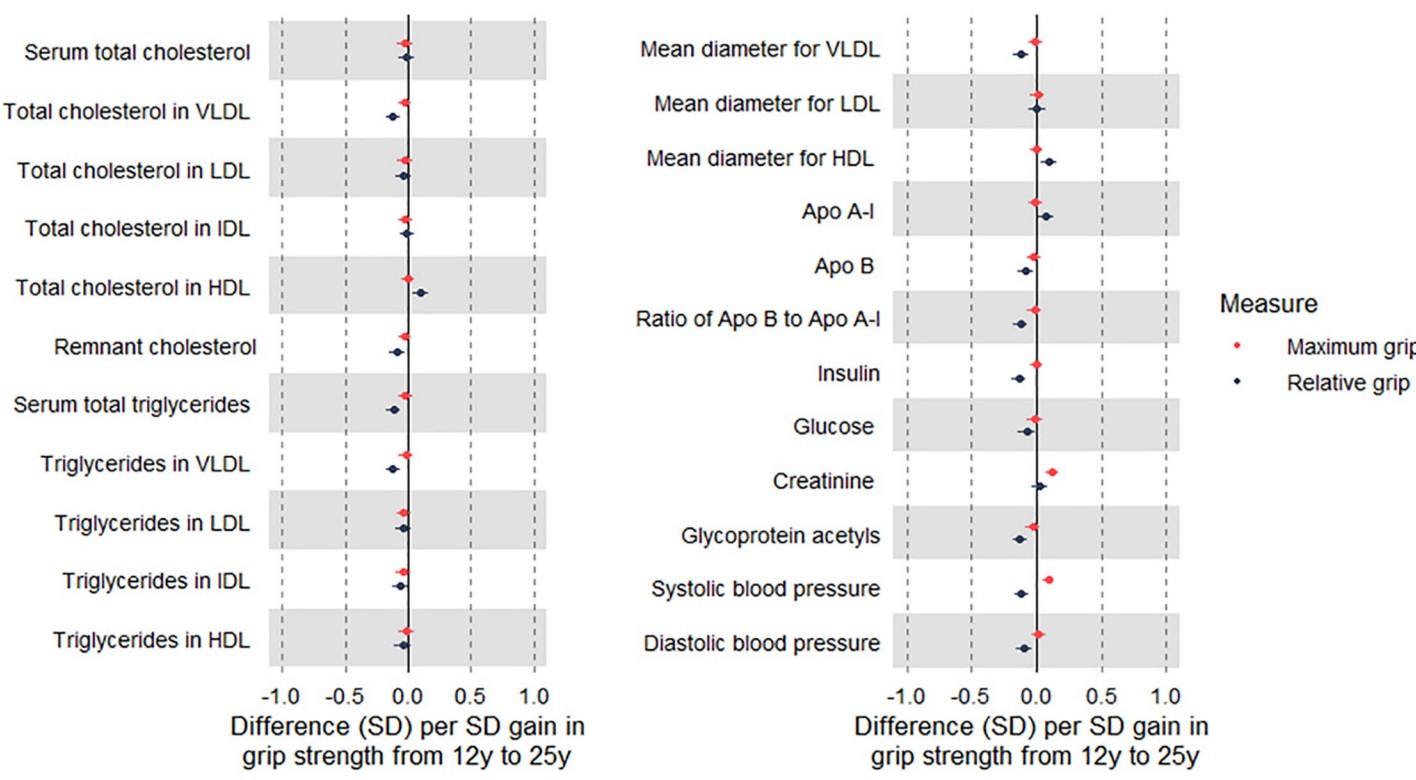

**Fig 5. Associations of change in grip strength from age 12 y to 25 y with cardiometabolic traits at 25 y among Avon Longitudinal Study of Parents and Children Generation 1 individuals.** Estimates are beta coefficients and 95% CIs representing SD-unit differences in cardiometabolic traits at 25 y per SD-unit change in grip strength (based on standardised grip at 25 y minus standardised grip at 12 y). Models are adjusted for age, sex, ethnicity, maternal education, grip strength at 12 y, and change in total fat mass index (except for relative grip models) (N range: 1,246 to 1,678). Maximum grip is based on maximum recorded grip strength of dominant hand (mean of 3 measures, in kilograms). Relative grip is based on maximum grip strength divided by weight. Apo, apolipoprotein; LDL, low-density lipoprotein; HDL, high-density lipoprotein; IDL, intermediate-density lipoprotein; VLDL, very-low-density lipoprotein.

particularly among males. Altogether, the results suggest that body muscle is less robustly associated with markers of CVD susceptibility than body fat and may therefore be a lower-priority intervention target.

The cardiometabolic traits considered here include several lipoproteins that previous ALSPAC analyses suggested are positively associated with a genetic risk score for adult CHD; higher levels of these traits are thus taken to reflect greater liability for developing CHD [49]. These liability traits/features, already apparent in childhood, include cholesterol and triglycerides within non-HDL particles that contain apolipoprotein B, which enables lipid-mediated atherosclerosis [50]. In the present study, evidence was very weak for associations of change in limb lean mass index with these lipid types. Gain in limb lean mass in adolescence was more strongly associated with lower apolipoprotein B lipids; this association appeared to be further limited to adolescent gains occurring within arms and was more pronounced among males, indicating that favourable associations of muscle gain may be sensitive to stage of body development, limb compartment, and sex. However, the apparent specificity for arms most likely reflects residual confounding of leg lean mass by ectopic fat; this is supported by stronger associations of arm lean mass change (versus leg) with higher creatinine (a muscle product/positive control), and by weaker correlations of arm lean mass change (versus leg) with fat mass change.

Results based on cross-sectional models of lean mass indices in relation to atherogenic traits were not consistent with results based on prospective models. When considering measures

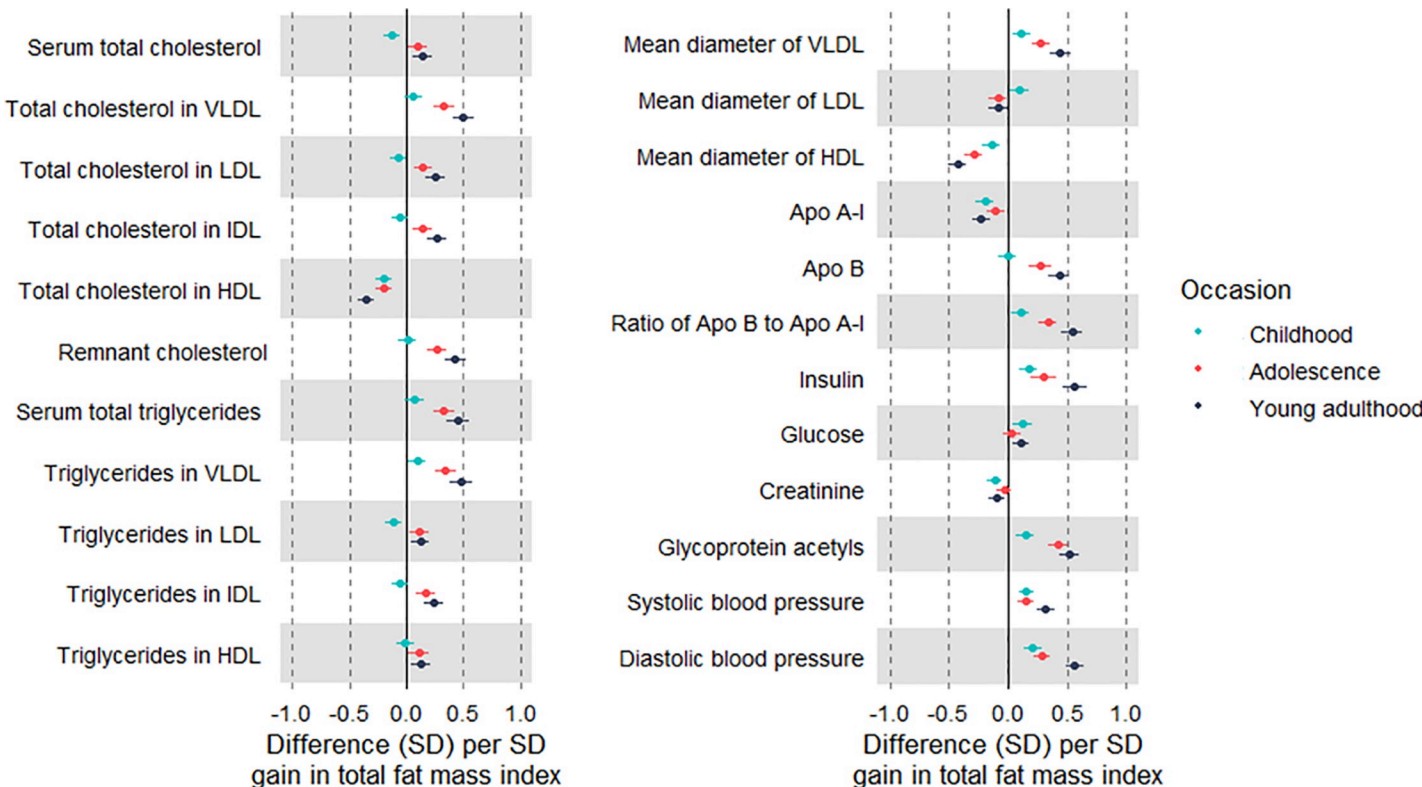

**Fig 6. Associations of change in fat mass index at different life stages with cardiometabolic traits at 25 y among Avon Longitudinal Study of Parents and Children Generation 1 individuals.** Estimates are beta coefficients and 95% CIs representing SD-unit differences in cardiometabolic traits at 25 y per SD-unit gain in total fat mass index in childhood (based on standardised index at 13 y minus standardised index at 10 y), adolescence (based on standardised index at 18 y minus standardised index at 13 y), and young adulthood (based on standardised index at 25 y minus standardised index at 18 y). Models adjusted for age, sex, ethnicity, maternal education, initial fat mass index, and change in limb lean mass index; models of adolescence additionally adjusted for puberty timing, and models for young adulthood additionally adjusted for puberty timing and smoking and alcohol at 18 y (*N* range: 1,532 to 2,804). Apo, apolipoprotein; LDL, low-density lipoprotein; HDL, high-density lipoprotein; IDL, intermediate-density lipoprotein; VLDL, very-low-density lipoprotein.

taken in young adulthood only, higher limb lean mass index was more weakly associated with lower atherogenic trait levels, and these associations appeared to be driven by higher lean mass held within legs; leg lean mass index was in turn more positively associated with creatinine than was arm lean mass index. This is in strong contrast to cross-sectional associations of higher fat mass index with atherogenic traits, which were highly consistent with prospective associations in terms of the direction and magnitude of point estimates. Thus, if reflective of causality, gaining muscle mass over time may be anticipated to confer greater cardiometabolic benefits than simply having higher current or usual levels of muscle mass; but with benefits likely of a lower magnitude than those anticipated for fat loss. Alternatively, the fragility of results for lean mass indices, as compared with fat mass index, could be more indicative of bias and residual confounding in estimates of lean mass than of robust causality.

Associations of muscle strengthening with atherogenic traits were highly dependent on the grip measure used. Change in grip measured in absolute units (maximum grip) was positively associated with creatinine but generally unassociated with atherogenic traits, with point estimates close to null values of no difference. In contrast, change in grip measured in relative units (as a function of body weight) was also positively associated with creatinine but generally only weakly associated with atherogenic traits in directions assumed to indicate better health, e.g., lower apolipoprotein B lipids, glycaemia, and inflammation. The magnitudes of these

associations of change in relative grip were higher than those seen for change in arm lean mass index but generally lower than those seen for change in fat mass index, particularly among males, where evidence was strongest. Furthermore, associations of higher maximum and relative grip strength with atherogenic traits were even less evident when assessed cross-sectionally in young adulthood, with point estimates close to null values. Considered alongside results on lean and fat mass indices, this supports body muscle as less robustly associated with cardiometabolic health in young adulthood than body fat, with a lower anticipated impact on CVD susceptibility from its modification. Resistance-based (muscle-building) physical activity is supported by several randomised controlled trials (RCTs) among adults with or without metabolic dysfunction as reducing blood pressure, but as having little to no effect on reducing glucose or non-HDL lipids [51,52]. Such trials are typically small ($N < 100$), with short follow-up ($<1$ y), but glycaemic and lipid benefits are seen in other RCTs of resistance- and aerobic-based activities among adults with type 2 diabetes [53,54]. Any physical activity biomechanically involves contracting some muscle, and habitual activity may mark contractile frequency. Prospective observational evidence supports favourable associations of habitual activity with cardiometabolic health [55,56], seen also among ALSPAC adolescents [57] at about half the magnitude presently seen for body fat. Altogether, evidence seems to indicate that the regular use of muscle matters more than the intentional building up of muscle for reducing CVD susceptibility.

Whether associations of lean mass gain with susceptibility traits are truly sensitive to body developmental stage is uncertain. Presently, associations of change in arm lean mass index (the compartment associated with higher creatinine and taken to best reflect muscle) with cardiometabolic traits were in directions assumed unfavourable to health for gains in childhood, favourable for gains in adolescence, and null for gains in young adulthood. Adolescence is an active period of growth and development following puberty [35], but how this may confer exclusive benefits of muscle gain is unclear. Several RCTs of resistance-based activity among children and adolescents support benefits for blood pressure, glycaemia, and lipids among both age groups [58], and there is suggestion of larger benefits with greater maturity [34], possibly reflecting greater modifiability of muscle. In the present study, grip was not measured in adolescence to enable comparisons, but associations seen for lean gain were distinct from those seen for fat gain, which involved consistently unfavourable associations with traits across sub-periods, with a tendency for larger effect sizes for more recent gains. Replication of associations in different study samples with triangulated approaches involving different sources of bias [59] is needed.

## Study limitations

This study is observational and effect estimates are prone to biases from unmeasured and poorly measured confounders. Major sources of confounding of the relationship between muscle and fat and cardiometabolic traits are expected to include subclinical disease and behaviours like smoking, but both of these factors are expected to be less influential at younger ages given their rarity or recency of onset. In addition to current exposure levels, our analyses were based on changes in body muscle measures occurring early in life in relation to cardiometabolic traits measured in young adulthood (age 25 y). These included an observation period of young adulthood (age 18 to 25 y), which should be relevant to adulthood more broadly, but results based on changes occurring over childhood and adolescence (periods of growth and development with pronounced hormonal activity) may not represent the impact of changes occurring later in life. Muscle mass and strength may plausibly be more influential for CVD susceptibility at older ages, when CVDs commonly start to emerge; obtaining unbiased

estimates of effect in middle to later adulthood is difficult, however, because of increased potential for residual confounding (reverse causation). Other study designs such as MR may prove useful here.

Muscle mass was estimated using DXA, which is more precise than bioimpedance but less granular than MRI and less able to exclude ectopic fat stored within muscle. We examined correlations among change in each limb lean mass compartment and total fat mass, which helps identify compartments that are more susceptible to confounding by residual fat. Grip was not measured in adolescence, preventing examination of change in grip across sub-periods. The participants analysed were relatively lean and predominantly white European; this limits generalisability to other groups but reduces confounding by disease and ancestral population structure. Sample sizes were modest; this is a tradeoff of detailed phenotyping and reduces precision. We applied metabolomics in a holistic manner; the large scope of analyses, particularly given sex differences, prevented examination of the shape of associations. Changes in exposures and outcomes are assumed to be linearly associated, i.e., negative change (loss) the inverse of positive change (gain); this may not always hold. Examinations of non-linearity are needed and would be aided by the prioritisation of cardiometabolic traits with triangulated evidence of causality for CVD.

## Conclusions

Our results suggest that muscle strengthening, as indicated by grip strength gain, is weakly associated with lower atherogenic trait levels in young adulthood, particularly among males. Such associations of gain in muscle mass with traits appear smaller and limited to gains occurring in adolescence. Gaining body fat was more consistently associated with the same atherogenic traits, in unfavourable directions and at larger magnitudes than seen for muscle mass or strength, again particularly among males. Altogether, results suggest that body muscle is less robustly associated with markers of CVD susceptibility than body fat and may therefore be a lower-priority intervention target.

## Supporting information

**S1 STROBE Checklist. Strengthening the Reporting of Observational Studies in Epidemiology (STROBE) checklist.**
(PDF)

**S1 Fig. Sex-specific distributions of lean and fat mass indices (kg/m$^2$) at age 10 y.**
(PDF)

**S2 Fig. Sex-specific distributions of lean and fat mass indices (kg/m$^2$) at age 13 y.**
(PDF)

**S3 Fig. Sex-specific distributions of lean and fat mass indices (kg/m$^2$) at age 18 y.**
(PDF)

**S4 Fig. Sex-specific distributions of lean and fat mass indices (kg/m$^2$) at age 25 y.**
(PDF)

**S5 Fig. Sex-specific distributions of maximum and relative grip strength in childhood and young adulthood.** Units of maximum grip are kilograms at age 12 y and pounds per square inch at age 25 y. Units of relative grip are kilograms/weight in kilograms at age 12 y and pounds per square inch/weight in kilograms at age 25 y.
(PDF)

**S6 Fig. Sex-specific changes in lean and fat mass indices from childhood to young adulthood.** Change values are based on difference scores (25-y value minus 10-y value), in original units (kg/m$^2$).
(PDF)

**S7 Fig. Sex-specific changes in lean and fat mass indices in adolescence.** Change values are based on difference scores (13-y value minus 10-y value), in original units (kg/m$^2$).
(PDF)

**S8 Fig. Sex-specific changes in lean and fat mass indices in adolescence.** Change values are based on difference scores (18-y value minus 13-y value), in original units (kg/m$^2$).
(PDF)

**S9 Fig. Sex-specific changes in lean and fat mass indices in young adulthood.** Change values are based on difference scores (25-y value minus 18-y value), in original units (kg/m$^2$).
(PDF)

**S10 Fig. Sex-specific changes in handgrip strength from childhood to young adulthood.** Change values are based on difference in SD units (25-y *z*-score value minus 12-y *z*-score value), given the different original measurement units between occasions.
(PDF)

**S1 Study Plan. Prospective study plan (ALSPAC data application).**
(PDF)

**S1 Table. Characteristics of 12,418 ALSPAC Generation 1 offspring not eligible for present analyses.** Values are mean (SD) unless otherwise noted. Described are those without data on each lean, grip, and fat measure on any occasion, at least 1 cardiometabolic trait at 25 y, and each covariate used for models. *Change is from age 12 y to 25 y and is based on difference in SD units given the different original measurement units between occasions.
(PDF)

**S2 Table. Pearson correlations between changes in limb lean mass indices and total fat mass index: 10 y to 25 y.**
(PDF)

**S3 Table. Pearson correlations among changes in limb lean mass indices, total fat mass index, and grip strength: 12 y to 25 y.**
(PDF)

**S4 Table. Pearson correlations between change in limb lean mass indices and total fat mass index: 10 y to 13 y.**
(PDF)

**S5 Table. Pearson correlations between change in limb lean mass indices and total fat mass index: 13 y to 18 y.**
(PDF)

**S6 Table. Pearson correlations between change in limb lean mass indices and total fat mass index: 18 y to 25 y.**
(PDF)

**S7 Table. Pearson correlations among limb lean mass indices, total fat mass index, and grip strength at 25 y.**
(PDF)

**S8 Table. Associations of change in lean mass indices from childhood to young adulthood with cardiometabolic traits in young adulthood among ALSPAC Generation 1 offspring.** (XLSX)

**S9 Table. Associations of change in lean mass indices from childhood to young adulthood with cardiometabolic traits in young adulthood among ALSPAC Generation 1 females.** (XLSX)

**S10 Table. Associations of change in lean mass indices from childhood to young adulthood with cardiometabolic traits in young adulthood among ALSPAC Generation 1 males.** (XLSX)

**S11 Table. Associations of change in lean mass indices in childhood with cardiometabolic traits in young adulthood among ALSPAC Generation 1 offspring.** (XLSX)

**S12 Table. Associations of change in lean mass indices in childhood with cardiometabolic traits in young adulthood among ALSPAC Generation 1 females.** (XLSX)

**S13 Table. Associations of change in lean mass indices in childhood with cardiometabolic traits in young adulthood among ALSPAC Generation 1 males.** (XLSX)

**S14 Table. Associations of change in lean mass indices in adolescence with cardiometabolic traits in young adulthood among ALSPAC Generation 1 offspring.** (XLSX)

**S15 Table. Associations of change in lean mass indices in adolescence with cardiometabolic traits in young adulthood among ALSPAC Generation 1 females.** (XLSX)

**S16 Table. Associations of change in lean mass indices in adolescence with cardiometabolic traits in young adulthood among ALSPAC Generation 1 males.** (XLSX)

**S17 Table. Associations of change in lean mass indices in young adulthood with cardiometabolic traits in young adulthood among ALSPAC Generation 1 offspring.** (XLSX)

**S18 Table. Associations of change in lean mass indices in young adulthood with cardiometabolic traits in young adulthood among ALSPAC Generation 1 females.** (XLSX)

**S19 Table. Associations of change in lean mass indices in young adulthood with cardiometabolic traits in young adulthood among ALSPAC Generation 1 males.** (XLSX)

**S20 Table. Associations of change in grip strength from childhood to young adulthood with cardiometabolic traits in young adulthood among ALSPAC Generation 1 offspring.** (XLSX)

**S21 Table. Associations of change in grip strength from childhood to young adulthood with cardiometabolic traits in young adulthood among ALSPAC Generation 1 females.** (XLSX)

**S22 Table. Associations of change in grip strength from childhood to young adulthood with cardiometabolic traits in young adulthood among ALSPAC Generation 1 males.** (XLSX)

**S23 Table. Associations of change in total fat mass index across different life stages with cardiometabolic traits in young adulthood among ALSPAC Generation 1 offspring.** (XLSX)

**S24 Table. Associations of change in total fat mass index across different life stages with cardiometabolic traits in young adulthood among ALSPAC Generation 1 females.** (XLSX)

**S25 Table. Associations of change in total fat mass index across different life stages with cardiometabolic traits in young adulthood among ALSPAC Generation 1 males.** (XLSX)

**S26 Table. Associations of lean mass indices with cardiometabolic traits in young adulthood among ALSPAC Generation 1 offspring.** (XLSX)

**S27 Table. Associations of lean mass indices with cardiometabolic traits in young adulthood among ALSPAC Generation 1 females.** (XLSX)

**S28 Table. Associations of lean mass indices with cardiometabolic traits in young adulthood among ALSPAC Generation 1 males.** (XLSX)

**S29 Table. Associations of grip strength with cardiometabolic traits in young adulthood among ALSPAC Generation 1 offspring.** (XLSX)

**S30 Table. Associations of grip strength with cardiometabolic traits in young adulthood among ALSPAC Generation 1 females.** (XLSX)

**S31 Table. Associations of grip strength with cardiometabolic traits in young adulthood among ALSPAC Generation 1 males.** (XLSX)

**S32 Table. Associations of total fat mass index with cardiometabolic traits in young adulthood among ALSPAC Generation 1 offspring.** (XLSX)

## Acknowledgments

We are extremely grateful to all the families who took part in this study, the midwives for their help in recruiting them, and the whole ALSPAC team, which includes interviewers, computer and laboratory technicians, clerical workers, research scientists, volunteers, managers, receptionists, and nurses.

## Author Contributions

**Conceptualization:** Joshua A. Bell.

**Formal analysis:** Joshua A. Bell.

**Investigation:** Joshua A. Bell, Kaitlin H. Wade, Linda M. O'Keeffe, David Carslake, Emma E. Vincent, Michael V. Holmes, Nicholas J. Timpson, George Davey Smith.

**Methodology:** Joshua A. Bell, Kaitlin H. Wade, Linda M. O'Keeffe, David Carslake, Emma E. Vincent, Michael V. Holmes, Nicholas J. Timpson, George Davey Smith.

**Supervision:** George Davey Smith.

**Visualization:** Joshua A. Bell.

**Writing – original draft:** Joshua A. Bell.

**Writing – review & editing:** Joshua A. Bell, Kaitlin H. Wade, Linda M. O'Keeffe, David Carslake, Emma E. Vincent, Michael V. Holmes, Nicholas J. Timpson, George Davey Smith.

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
