## [Editor Report · Decision Letter 0]

17 Sep 2020

Dear Dr Bell, 

Thank you for submitting your manuscript entitled "Body muscle gain and markers of cardiovascular disease susceptibility in young adulthood: prospective cohort study" for consideration by PLOS Medicine.

Your manuscript has now been evaluated by the PLOS Medicine editorial staff and I am writing to let you know that we would like to send your submission out for external peer review.

Kind regards,

Helen Howard, for Clare Stone PhD 

Acting Editor-in-Chief

PLOS Medicine 

plosmedicine.org

---

## [Decision Letter · Decision Letter 1]

2 Nov 2020

Dear Dr. Bell,

Thank you very much for submitting your manuscript "Body muscle gain and markers of cardiovascular disease susceptibility in young adulthood: prospective cohort study" (PMEDICINE-D-20-04485R1) for consideration at PLOS Medicine. 

Your paper was evaluated by a senior editor and discussed among all the editors here. It was also evaluated by three independent reviewers, including a statistical reviewer (r#1). The reviews are appended at the bottom of this email and any accompanying reviewer attachments can be seen via the link below:

[LINK]

In light of these reviews, I am afraid that we will not be able to accept the manuscript for publication in the journal in its current form, but we would like to consider a revised version that addresses the reviewers' and editors' comments. Obviously we cannot make any decision about publication until we have seen the revised manuscript and your response, and we plan to seek re-review by one or more of the reviewers. 

We expect to receive your revised manuscript by Nov 19 2020 11:59PM. Please email us (plosmedicine@plos.org) if you have any questions or concerns.

We look forward to receiving your revised manuscript. 

Sincerely,

Adya Misra, PhD

Senior Editor 

PLOS Medicine

plosmedicine.org

*At this stage, we ask that you include a short, non-technical Author Summary of your research to make findings accessible to a wide audience that includes both scientists and non-scientists. The Author Summary should immediately follow the Abstract in your revised manuscript. This text is subject to editorial change and should be distinct from the scientific abstract. Please see our author guidelines for more information: https://journals.plos.org/plosmedicine/s/revising-your-manuscript#loc-author-summary

*Please clarify if the analytical approach reported here corresponded to one laid out in a prospective protocol or analysis plan? Please state this (either way) early in the Methods section.

*We'd suggest ensuring that the study is reported according to the STROBE guideline, and the completed STROBE checklist should be included as Supporting Information with the revised paper. Please add the following statement, or similar, to the Methods: "This study is reported as per the Strengthening the Reporting of Observational Studies in Epidemiology (STROBE) guideline (S1 Checklist)." The STROBE guideline can be found here: http://www.equator-network.org/reporting-guidelines/strobe/. When completing the checklist, please use section and paragraph numbers, rather than page numbers.

Comments from the reviewers:

Reviewer #1: I confine my remarks to statistical aspects of this paper. Unfortunately, I think some major changes need to be made.

The biggest is that, rather than look at change in the dependent variables, the authors should use either a multilevel model or generalized estimating equations to deal with the dependent errors. 

Other issues

p. 3 "strong cause" .... has causation been estatblished here?

 why look at total lean mass? Surely someone who is 2 meters tall will have a lot more mass than someone who is 1.5 m. tall. 

 Handgrip as an indicator of overall strength seems a bit problematic too. Handgrip only tests strength in one muscle.

 "No protection" do you mean no significant protection? Give effect sizes. 

p. 6 

 Lumping all nonwhites together seems arbitrary and likely to increase error.

 Independent variables like alcohol consumption should not be categorized. See Frank Harrell "Regression modeling strategies"

Peter Flom

Reviewer #2: It is an elegant and exhaustive paper that shows little added value to consider muscle / lean mass trajectories during childhood, compared to fat mass, as a risk factor for atherogenic profile in young adulthood. The research question is nicely introduced, and the analyses seem thorough. I have two main concerns regarding the analyses that will need to be addressed though: 1) account for multiple testing; 2) unrestricted sample analysis is not a clean way to compare estimates as the samples are varying from trait to trait / exposure to exposure. Also overall the conclusions are not supported by the results and should be toned down.

My specific comments are detailed below:

Abstract 

- "and others" should be removed or specified.

- "further specific"?

- Conclusion first sentence should be rephrased.

Intro 

- "… while Mendelian randomization estimates suggest no protection from stronger grip independent of BMI (16)." Here you probably need to highlight that there is no Mendelian randomization on limb lean mass?

- " Males have higher lean mass and stronger grip than females (13, 17), but whether these are more cardiometabolically beneficial among males is unknown." This sentence is somewhat paradoxical. Males at higher risk of CVD/CHD than women, so having more lean mass does not appear to be potentially protective. And why would lean mass / strength be more cardiometabolically beneficial among males than among women? You are talking about strength of association? This justification of the exploration of the interaction with sex is a bit poor and would need to be developed. 

- 

Throughout the manuscript: every time you use "these traits" it is somewhat imprecise, I think it would be better to use "atherogenic traits" throughout.

Methods

- Confounder: ideally, alcohol and smoking would be measured at time of exposure. Was this question not asked at earlier visits than at 25y? 

- I have concerns on the unrestricted sample analysis with N varying, as the sample selection makes for each trait / time period is likely to making results not comparable. I suggest the complete case be the main analysis, at least with available data at 10y and 25y.

- Did you test for interactions with sex if you stratified your results by sex and seem to have a strong hypothesis that the associations might be different for males vs females

- If you looked at 145 concentrations and 79 ratios of metabolomics markers, you should account for multiple testing in some way. It is unclear if you tested all of the markers of if you did some preselection.

Results: 

- Ineligible vs included participants. Nowhere in the Method section is the explanation of who are the ineligible participants. Please insert the information from the flow diagram Figure 1 into the Method section.

- The paragraph "Correlations amongst changes in lean mass indices, grip strength, and fat mass index" does not flow easily, please try and rephrase. 

- Page 10 Associations of change in grip strength with cardiometabolic traits Please state in the sentence the direction of the association with VLDL triglycerides

Discussion

The summary page 11"suggest that muscle strengthening is associated with lower atherogenic trait levels" (also found in the abstract) does not reflect accurately the results, as the associations of change in limb lean mass from 10y to 25y with most atherogenic traits were of modest magnitude and mostly in directions assumed to reflect poorer health. Please tone down this sentence. This goes for the last paragraph of the discussion and for the abstract as well.

- Page 12, I do not understand the phrase "phenotypic features of genetically liability to adult CHD". 

- Page 12. I also do not understand the logic in the last sentence of the paragraph "Change in leg lean mass may also better reflect body shape and abdominal fat storage, the compartment driving effects of total fat". Are you saying that leg lean mass gain is a proxy for total fat gain? Why is that? Can you please expand? 

- Page 12. The following paragraph is somewhat empty. I think one could say that the fact that the relative index is more strongly associated may be reflecting simply the strong association with fat mass? 

Reviewer #3: The role of muscle mass and muscular strength in risk of cardiovascular and metabolic disease is gaining increasing interest. This study aimed to determine the associations between changes in these variables (as well as changes in adipsosity variables) from adolescences to young adulthood and cardiometabolic traits at age 25. The paper is well written and the analyses have been comprehsively undertaken. I do however have a few concerns that the authors should address:

Major

1) It is unclear why only change in grip strength, lean mass and fat mass between adolescence and young adulthood was considered here, and not the actual values for these variables at age 25. Changes occuring over the course of maturation and puberty may not have the same effect as changes in these variables over the course of adulthood, so interpretation of these findings as to the importance of lean mass and muscular strength in affacting cardiometabolic traits is uncertain. The authors need to provide more justification about what assessing the change in these variables over time adds over and above just considering these variables at age 25. They should comment on whether you can extrapolate these changes to to other stages of life in the discussion. Could the authors also undertake the analyses using the values of grip strength, lean mass and fat mass variables at age 25 as the predictor variables? It would helpful to compare the results of these two analyses and comment as approriate in the discussion.

2) It is unclear why the authors chose to express grip strength relative to fat mass index in particular. Please justify this. Why not relative to body weight, or height, or BMI or lean mass index? All of these relative indices have limitations and the authors need to be clear why they have chosen to express the findings in this way. Some of the results where there was an inverse relationship between relative but not absolute grip stength and atherogenic traits (e.g. for VLDL trigycleride) is probably driven by the the inverse relationship between fat mass and VLDL-TG, rather than the association with grip strength per se. The authors should consider this carefully and perhaps express relative grip strength relative to body weight or height. There are publications which have addressed the best approaches to normalise grip strength and the authors would be advised to consult these and justify their approach here.

3) On page 4, the authors state that the causality of body muscle for CVD risk can be interogated by examining the assocations with key intermediates. I don't agree with this. Although the analyses will show correlations between muscle/strength and risk traits, it is entirely possible that this could be driven by confounders in this type of analysis. This statement should be tempered.

Minor

1) Abstract, page 2, line 6. This sentence is not clear - is there a word or a comma missing?

2) Page 3, four lines up from bottom. A handgrip test provides a direct measurement of grip strength, so it is not correct to say that this in an indirect measure. You can say that grip strength is highly correlated with strength in other muscle groups so can provide a reasonable proxy for overall muscular strength.

2) Page 4, lines 1 and 2. This sentence is unclear. Why would you expect a sex difference here? Is this an example of HARKing?

[LINK]

---

## [Decision Letter · Decision Letter 2]

25 Apr 2021

Dear Dr. Bell,

Thank you very much for submitting your revised manuscript "Body muscle gain and markers of cardiovascular disease susceptibility in young adulthood: prospective cohort study" (PMEDICINE-D-20-04485R2) for consideration at PLOS Medicine. We do apologize for the delay in sending you a response. 

Your paper was re-evaluated by our reviewers, including a statistical reviewer. The reviews are appended at the bottom of this email and any accompanying reviewer attachments can be seen via the link below:

[LINK]

In light of these reviews, we will be unable to accept the manuscript for publication in the journal in its current form, but we would like to invite you to submit a further revised version that addresses the reviewers' and editors' comments. You will recognize that we cannot make a decision about publication until we have seen the revised manuscript and your response, and we may seek re-review by one or more of the reviewers. 

We hope to receive your revised manuscript by May 14 2021 11:59PM. Please email us (plosmedicine@plos.org) if you have any questions or concerns.

Please let me know if you have any questions, and we look forward to receiving your revised manuscript. 

Sincerely,

Richard Turner, PhD

rturner@plos.org

Please remove the word "prospective" from the title, and adapt the study descriptor to "A cohort study".

Please quote study dates in the abstract.

Please restructure the author summary to consist of three subsections, each of 3-4 short points (1-2 short sentences each). 

Please specify "Written informed consent" in the Methods section, assuming this is correct.

Please spell out "years" throughout the text.

Throughout the article, please adapt the reference call-outs to the following style: "... and glucose [6,7]." (noting the absence of spaces within the square brackets). 

Please remove the information on funding, competing interests and data access from the end of the main text. In the event of publication, this will appear in the article metadata, via entries in the submission form. 

Please make that "PLoS ONE" in the reference list. 

Please add "[preprint]" to reference 30 and any other preprints cited. 

Noting references 41 & 48, please ensure that all citations have full access information. 

Please rename the STROBE checklist "S2_STROBE_Checkist", or similar, and refer to it as such in the Methods section. 

Comments from the reviewers:

*** Reviewer #1: 

The authors have addressed my concerns and I now recommend publication

Peter Flom

*** Reviewer #2: 

Thank you, the authors have responded very satisfactorily to my comments and have improved the manuscript. I have no further comment.

*** Reviewer #3: 

Thank you for your comprehensive response to my queries. I appreciate the considerable extra work you have done and think that the paper is very much improved as a result. I do however still have a couple of substantive concerns which I have only noticed when reading through the revised paper (sorry that I missed these first time around) and a few minor comments as follows:

Major

1) Looking through the list of covariates adjusted for, I notice that baseline values for the outcome of interest were not included. I think that it would be helpful to add this into the models. In particular it seems that the cross-sectional assocation between limb lean mass index and atherogenic traits, while weak, are directionally different from the assocations between changes in limb lean mass index and athergenic traits. I wonder if this lack of adjustment for baseline values in the atherogenic traits contributes?

2) Values for grip strength appeared to decrease on average between age 12 and 25. This is very surprising, particularly in men - we would expect a 25 year old man to be considerably stronger than a 12 year old boy. Values for grip strength at age 25 are also vey low - less than half the level of that in UK Biobank in adults aged 40-70 years, who you would expect to be less strong than 25 year olds. Indeed, mean values are in the sarcopenic range. If there are errors in this measurement, this would call the entire analysis related to grip strength in this manuscript into question. An initial thought is that you might have inadvertently switched the values for 12 vs 25 years, but something else may be going on? Please could you recheck your data. It is not credible for grip strength amongst healthy 25 year olds to be this low.

Minor

1) Table 1, please could you also report data for males and females separately. This is important for values such as grip strength and fat and lean mass indexes which will differ substantially according to sex.

2) I am not sure that I fully agree with this statement in the "Why was this study done?" section: "Examining naturally occurring changes in lean mass, grip strength, and fat mass across early stages of life, when ageing-related chronic diseases are rare, should naturally reduce the potential for reverse causation and enable less biased estimates of effect of each body compartment on markers of heart health.". I would accept this argument if you were comparing changes from say 18 to 30, but not so much for changes from 12 to 25, when developmental changes are likely to play a substantial confounding role that you cannot fully adjust for. Similarly, I am not sure I fully agree with the earlier comment "This enabled us to compare how strongly different body compartments relate to heart health, and to pinpoint at what life stage body fat and muscle may most impact on health." You did not look at several life stages here - just adolescence - so both these statements need to be tempered.

***

[LINK]

---

## [Decision Letter · Decision Letter 3]

31 Jul 2021

Dear Dr. Bell,

Thank you very much for re-submitting your manuscript "Body muscle gain and markers of cardiovascular disease susceptibility in young adulthood: A cohort study" (PMEDICINE-D-20-04485R3) for consideration at PLOS Medicine. We apologize for the delay in sending you a response. 

I have discussed the paper with editorial colleagues and it was also seen again by one reviewer. I am pleased to tell you that, provided the remaining editorial and production issues are fully dealt with, with we expect to be able to accept the paper for publication in the journal.

[LINK]

Please let me know if you have any questions, and we look forward to receiving the revised manuscript.   

Sincerely,

Richard Turner, PhD

rturner@plos.org

Requests from Editors:

Please add a few words to your data statement (submission form) to explain the reasons that data cannot be made openly available, e.g., "Owing to constraints of the study's ethics approval, individual-level ALSPAC data are available upon application only ...". 

Please adapt the "Conclusions" subsection of your abstract to begin: "In this study, we found that ... was weakly associated ..." or similar. 

Please avoid the long bulleted point in the second subsection of your Author summary. This could be split into two points, perhaps.

Comments from Reviewers:

*** Reviewer #3: 

The authors have done a very good job in addressing my concerns. The use of two different devices to measure grip strength is not ideal, but I am happy with the way that this has been addressed and explained in the text. Happy to accept now.

***

[LINK]

---

## [Editor Report · Decision Letter 4]

3 Aug 2021

Dear Dr Bell, 

On behalf of my colleagues and the Academic Editor, Dr Gill, I am pleased to inform you that we have agreed to publish your manuscript "Body muscle gain and markers of cardiovascular disease susceptibility in young adulthood: A cohort study" (PMEDICINE-D-20-04485R4) in PLOS Medicine.

PRESS

Sincerely, 

Richard Turner, PhD 

rturner@plos.org